# Genome-Wide Identification and Expression Analysis of MIKC-Type MADS-Box Gene Family in *Punica granatum* L.

**Yujie Zhao** [1,2]**, Honglian Zhao** [1,2]**, Yuying Wang** [1,2]**, Xinhui Zhang** [1,2]**, Xueqing Zhao** [1,2] **and Zhaohe Yuan** [1,2,*]

[1]  Co-Innovation Center for Sustainable Forestry in Southern China, Nanjing Forestry University, Nanjing 210037, China; z1184985369@njfu.edu.cn (Y.Z.); honglian_zhao@hotmail.com (H.Z.); wangyuying@njfu.edu.cn (Y.W.); zhxinhui@hotmail.com (X.Z.); zhaoxq402@njfu.edu.cn (X.Z.)

[2]  College of Forestry, Nanjing Forestry University, Nanjing 210037, China

*  Correspondence: zhyuan88@hotmail.com

**Abstract:** MADS-box is a critical transcription factor regulating the development of floral organs and plays essential roles in the growth and development of floral transformation, flower meristem determination, the development of male and female gametophytes, and fruit development. In this study, 36 MIKC-type MADS-box genes were identified in the 'Taishanhong' pomegranate genome. By utilizing phylogenetic analysis, 36 genes were divided into 14 subfamilies. Bioinformatics methods were used to analyze the gene structure, conserved motifs, *cis*-acting elements, and the protein interaction networks of the MIKC-type MADS-box family members in pomegranate, and their expressions pattern in different tissues of pomegranate were analyzed. Tissue-specific expression analysis revealed that the E-class genes (*PgMADS03*, *PgMADS21,* and *PgMADS27*) were highly expressed in floral tissues, while *PgMADS29* was not expressed in all tissues, indicating that the functions of the E-class genes were differentiated. *PgMADS15* of the C/D-class was the key gene in the development network of pomegranate flower organs, suggesting that *PgMADS15* might play an essential role in the peel and inner seed coat development of pomegranate. The results in this study will provide a reference for the classification, cloning, and functional research of pomegranate MADS-box genes.

**Keywords:** Pomegranate; MIKC-type MADS-box genes; gene family identification; proteins interaction; expression analysis

## 1. Introduction

MADS-box (its name from Minichromosome maintenance 1 (MCM1), AGAMOUS (AG), DEFICIENS (DEF) and serum response factor (SRF)) genes encode transcription factors, have been widely found in plants, fungi, and animals, and contain the highly conservative MADS-domain composed of approximately 55 amino acids [1,2]. On the basis of genetic structures and phylogenetic analysis, the MADS-box family can be divided into two phylogenetically distinct groups: type I and type II [3]. Most of the well-studied plant genes are type II genes that have three more domains than those of type I genes: a more conservative intervening domain (I-domain), a moderately conservative keratin-like coiled-coil domain (K-domain), and a variable C-terminal domain (C-domain) [4]. These genes are considered as the MIKC-type and are specific to plants.

The plant-specific MIKC-type MADS-box genes were first identified as floral organ determinant genes in *Arabidopsis thaliana* and *Antirrhinum majus* [5,6]. MADS-box family genes are involved

in plant growth and development, flower transformation, flower meristem decision, male/female gametophyte development, fruit development and maturation, and somatic embryogenesis [7–10]. Type I only contain MADS-domains without K-domains and are called M-type [11]. Type I genes are divided into four subfamilies: Mα, Mβ, Mγ, and Mδ. Based on some research, the Mδ subfamily is referred to as type II, so it is also called MIKC*-type [12,13]. Type II genes can be divided into two subfamilies: MIKC* and MIKC$^C$. MIKC$^C$ subfamily can be divided into 13 branches: AG, AGL6 (AGAMOUS-LIKE, AGL), AGL12, AGL15, AGL17, AP1(APETALA, AP)/FUL (FRUITFULL), AP3/PI (PISTILLATA), FLC (FLOWERING LOCUS C), SOC1 (SUPPRESSOR OF OVEREXPRESSION OF CONSTANS1), SEP (SEPALLATA), SVP (SHORT VEGETATIVE PHASE), BS (B-SISTER), and TM8 [14,15]. In *Arabidopsis*, MIKC$^C$-type embraces 12 subfamilies, without Tomato MADS 8 (TM8). So, there is little research on TM8's function [16].

According to the ABCDE model of flower development, the five class genes of A, B, C, D, and E coordinately regulate the development of sepals, petals, stamens, carpels, and ovules. Arabidopsis has two A-class genes (*AP1* and *AP2*), two B-class genes (*PI* and *AP3*), C/D-class genes (*AG*, *AGL11* and *STK* (*SEEDSTICK*)), and a single E-class gene (*SEP*), of which only *AP2* is not a MADS-box gene [17]. The model suggests that A + E control meristem and sepals' formation, A + B + E regulates petals formation, B + C + E controls stamens formation, C + E controls carpels formation, and C + D + E regulates ovules development [17]. Martín-Pizarro et al. identified two B-class genes *FaAP3* and *FaTM6* in strawberries (*Fragaria × ananassa*) and used CRISPR/Cas9 technology for the first time to study the function of *FaTM6* in flower development. They found that *FaTM6* played a key role in the development of strawberry petals and anthers; the expression pattern of *FaTM6* was similar to *AP3* in *Arabidopsis* [18]. *AG* is the first reported MADS-box gene to regulate stamens and carpels development [19]. C-class genes regulate the establishment of floral meristems by inhibiting the activity of the A-class gene [20]. In the control carpels, *STK*, *SHP1*, and *SHP2* exist functional redundancy in some extent. *SHP1*, *SHP2*, and *STK* jointly regulate the development of ovules, with *SHP1/SHP2* sharing some function of C- and D-class [21]. Ehlers et al. found that *shp1* and *shp2* could not coordinate the development of ovules, indicating that *SHP1* and *SHP2* might be the key genes regulating ovules development [22]. There are two AG subfamily genes *TAG1* and *TAGL1* in tomato. Pan et al. found that the stamens and carpel of *TAGL1* RNAi strain were normal, and the fruit ripening was inhibited, speculating that *TAGL1* shows new functions [23]. The development of stamens and carpel in *TAG1* RNAi were significantly inhibited, indicating that *TAG1* plays a major the role in regulating the development of stamens and carpel [23]. Xu et al. cloned three *SEP*-like genes, *PrpMADS2*, *PrpMADS5*, and *PrpMADS7*, in peach [24]. They found that the transgenic plant of *PrpMADS2* had a similar phenotype with the wild. A strain with an overexpressed *PrpMADS5* flowered earlier than the control plant, and it was involved in the differentiation of flower primordia. *PrpMADS7* transgenic plants showed weak growth potential and earlier flowering than the wild, indicating new functions of the *SEP-like* gene [24]. SEP (SEPALLATA) participates in the formation of tetrameric protein complexes, and forms the complexes with A, B, and C-type proteins [25,26]. These complexes help to transform leaf primordia into floral organ primordium, of which ectopic expression of A, B, and E leads to the transformation of leaves into petals. The expression of B, C, and E gene causes the transformation of leaves into stamen organs [27].

Pomegranate (*Punica granatum* L.), belonging to Lythraceae family, is an excellent fruit tree with economic, nutritional, medicinal, ornamental, and ecological values [28]. Pomegranate is mostly cultivated in tropics and subtropics regions, such as the Mediterranean basin, Asia, Australia, and North America [29]. In China, the total scale of pomegranate cultivation is about 128,000 hm$^2$, with an annual output of about 1.7 million tons [28]. Pomegranate contains abundant polyphenols, tannins, anthocyanins, vitamins, and minerals, that are responsible to reduce blood pressure, and act against serious diseases such as cancer [30–32]. These have led to an increasing demand for the consumption of fresh fruit, juice, tea, and other pomegranate products. The completion genome sequencing of pomegranate and the publication of data [33–35] provided necessary data support for the promotion of

research on gene function of pomegranate. In this study, we aim to identify the MIKC-type MADS-box members of pomegranate by bioinformatics methods, trying to analyze their gene structure, *cis*-acting elements, and tissue-specific expression to explore their functions in the development of pomegranate flowers. The results of this study would lay a foundation for the cloning and functional research of *PgMADS* genes.

## 2. Materials and Methods

### 2.1. Data Collection for Expression Analysis

The transcription data of pomegranate were obtained from the NCBI database (https://www.ncbi.nlm.nih.gov/), including leaves, roots, bisexual flowers, functional male flowers, exocarp, pericarp and other tissues. The respective accession sequence numbers are 'Dabenzi' SRR5678820, SRR5279388 - SRR5279397; 'Baiyushizi' SRR5678819; 'Tunisian' SRR5446590 - SRR5446607; 'Black127' SRR1054190; 'Nana' SRR1055290; and 'Wonderful' SRR080723 (Table S1) [33,36–38].

MIKC-type MADS-box family protein sequences of *Arabidopsis thaliana* were downloaded from The Arabidopsis Information Source (https://www.arabidopsis.org/), and that of poplar (*Populus trichocarpa*) and apple (*Malus domestica*) were downloaded from Plant Transcription Factor Database (http://planttfdb.cbi.pku.edu.cn/index.php).

### 2.2. Identification and Sequence Analysis of Pomegranate MIKC-Type MADS-Box Transcription Factors

The HMM (Hidden Markov Models) profile of the Pfam MADS-domain (PF00319) was downloaded from the pfam database (http://pfam.xfam.org/). Then, the HMM profile was performed against pomegranate protein databases (GCA_002864125) using the HMMER 3.0 software package (The European Bionfirmatics Institute, Hinxton, Cambridgeshire, UK), (E-value < 1e$^{-5}$), and redundant sequences were removed manually. Moreover, all obtained MADS protein sequences were further analyzed on the CDD website (https://www.ncbi.nlm.nih.gov/cdd) [39] to verify the presence of the MADS-domain. Identified MADS-box protein sequence were categorized using the PlantTFDB to follow-up analysis for M-type and MIKC-type MADS-box members [40,41].

The published MADS-box protein sequences of Arabidopsis, poplar and apple were used as queries to perform BLAST against the pomegranate protein database (E-value < 1e$^{-10}$, identity > 50%), duplication was removed. Having predicted and divided members into MIKC-type and M-type using the PlantTFDB website [40,41], MIKC-type MADS-box protein sequences were screened and selected.

ExPASy Proteomics Server (https://web.expasy.org/protparam/) was used to predict the physicochemical properties of MIKC-type MADS-box protein, including amino acid sequence length, molecular weight, and isoelectric point.

### 2.3. Phylogenetic Analysis of Pomegranate MIKC-Type MADS-Box Family

Multiple sequences alignment of pomegranate, Arabidopsis (45 members) and Poplar (51 members) MIKC-type MADS-box proteins were carried out using ClustalX 2.1 software (http://www.clustal.org/omega/), and artificially corrected them. An unrooted neighbor-joining (NJ) phylogenetic tree was constructed with all of the MIKC-type MADS-box protein sequences from pomegranate, Arabidopsis, poplar using MEGA 7.0 (Penn State-A, State College, PA, USA) [42]. Bootstrap analysis was performed using 1000 repetitions with the following parameters: complete deletion and Poisson model. The phylogenetic tree was beautified with EvolView (https://www.evolgenius.info/evolview/#login).

The subfamily was classified according to the support rate of subfamily branches (>70) and the structural integrity of phylogenetic tree. Subfamily classification were obtained by according to Díaz-Riquelme et al. [15].

*2.4. Gene Structure Analysis and Motif Identification of Pomegranate MIKC-Type MADS-Box Family*

The protein sequences of the MIKC-type MADS-box were extracted from the pomegranate genome. Motifs of MIKC-type MADS-box proteins were identified using MEME online tools (http://meme-suite.org/tools/meme) with a default parameter, and the motif characteristics of the MIKC-type MADS-box proteins were obtained. The SMART program (http://smart.embl.de/) was used to further analysis for the conservative motifs.

The annotation information of 36 MIKC-type MADS-box genes was extracted from the pomegranate genome annotation file, then the results were submitted to Gene Structure Display Server (GSDS 2.0: http://gsds.cbi.pku.edu.cn) for online genetic structure analysis and drawing Schematic diagram of gene structures.

*2.5. Prediction Promoter Elements and Protein Interaction Network of Pomegranate MIKC-Type MADS-Box Family*

To identify putative *cis*-acting elements in promoters, 1500 bp upstream sequences of the initiation codon were obtained from the pomegranate genome sequence by TBtools (GitHub Inc., San Francisco, CA, USA) [43]. The *cis*-acting elements were analyzed by PlantCARE (http://bioinformatics.psb.ugent.be/webtools/plantcare/html/) [40]. Then, the result was presented as images using Tbtools.

The protein–protein interaction network of the MIKC-type MADS-box family was analyzed by String (https://string-db.org/).

*2.6. Expression Analysis for Pomegranate MIKC-Type MADS-Box Family*

To study the expression patterns of MIKC-type MADS-box genes in pomegranate different tissues, the published transcriptome data were downloaded as the primary data source for expression analysis from the NCBI database (http://www.ncbi.nlm.nih.gov/). Firstly, all RNA-Seq data were qualitatively controlled by fastp [44] to obtain cleaned reads. Then the sequences were indexed with pomegranate transcriptome data. Through the use of Kallisto 0.44.0 software (https://pachterlab.github.io/kallisto/download.html) [45], and the gene expression levels was further calculated and analyzed. The corresponding expression level (TPM (Transcripts Per Million) value) of MIKC-type MADS-box genes was thus obtained. TPM value was transformed into $Log_2$ (TPM + 1). Finally, the thermal map was drawn with Tbtools.

## 3. Results

*3.1. Identification and Sequence Analysis of Pomegranate MIKC-Type MADS-Box Family*

A total of 81 MADS-box family members were identified from the 'Taishanhong' pomegranate genome using the hmmsearch program. Through CDD online verification, all members contained the MADS-domain. After classifying of M-type and MIKC-Type family on PlantTFDB, 36 MIKC-type MADS-box candidate members were obtained.

Subsequently, 71 MADS-box family members were identified using the local blast (ftp://ftp.ncbi.nlm.nih.gov/blast/executables/blast+/LATEST/). It was found that eight members contained the type II specific K-domain, without MADS-domain, so they were discarded. Through classification analysis on PlantTFDB, 36 MIKC-type MADS-box members were obtained.

Based on the identification and classification of above two methods, a total of 36 MIKC-type MADS-box genes were identified in pomegranate, which were renamed as *PgMADS01~PgMADS36* according to the gene sequence number [13], for convenience in subsequent analysis (Table 1).

**Table 1.** The basic information of the pomegranate MIKC-type MADS-box gene family

| Gene Name | Gene ID | Location | CDS (bp) | Exon No. | Strand | Amino Acid Residues | MW(Da) | PI | Subfamily |
|---|---|---|---|---|---|---|---|---|---|
| *PgMADS01* | Pg000170.1 | scaffold1:4730801:4734403 | 720 | 7 | − | 239 | 27591.1 | 9.76 | A(AP1-FUL) |
| *PgMADS02* | Pg001270.1 | scaffold10:781136:787463 | 576 | 5 | − | 191 | 22251.1 | 8.76 | SOC1 |
| *PgMADS03* | Pg002695.1 | scaffold11:837228:841996 | 846 | 7 | − | 281 | 31958.1 | 8.19 | E(SEP) |
| *PgMADS04* | Pg002696.1 | scaffold11:827349:833260 | 600 | 7 | − | 199 | 22764.8 | 9.58 | FLC |
| *PgMADS05* | Pg002697.1 | scaffold11:821950:824737 | 603 | 6 | − | 200 | 22793.1 | 10.17 | FLC |
| *PgMADS06* | Pg003192.1 | scaffold111:252873:258483 | 522 | 6 | − | 173 | 20047.1 | 9.59 | C/D(AG) |
| *PgMADS07* | Pg003228.1 | scaffold111:423542:426294 | 645 | 7 | + | 214 | 24079.5 | 8.95 | AGL12 |
| *PgMADS08* | Pg003757.1 | scaffold119:139771:142418 | 639 | 7 | − | 212 | 24900.2 | 9.28 | B(AP3-PI) |
| *PgMADS09* | Pg005184.1 | scaffold13:3171597:3176055 | 741 | 8 | − | 246 | 28261 | 8.58 | E(SEP) |
| *PgMADS10* | Pg005185.1 | scaffold13:3153310:3158145 | 807 | 8 | − | 268 | 30475.7 | 8.8 | A(AP1-FUL) |
| *PgMADS11* | Pg006006.1 | scaffold136:390163:393471 | 906 | 9 | − | 301 | 34973.4 | 9.9 | BS |
| *PgMADS12* | Pg008155.1 | scaffold16:570614:573248 | 885 | 7 | − | 294 | 33598.8 | 6.39 | AGL15 |
| *PgMADS13* | Pg008465.1 | scaffold16:2554235:2556059 | 642 | 7 | + | 213 | 23721.7 | 8.48 | SVP |
| *PgMADS14* | Pg008776.1 | scaffold17:2387895:2392008 | 687 | 8 | − | 228 | 25779.9 | 5.52 | SVP |
| *PgMADS15* | Pg009904.1 | scaffold184:267398:273943 | 765 | 5 | − | 254 | 28775.2 | 8.88 | C/D(AG) |
| *PgMADS16* | Pg014226.1 | scaffold24:1271362:1277266 | 504 | 5 | + | 167 | 19349.0 | 5.97 | MIKC* |
| *PgMADS17* | Pg014283.1 | scaffold24:2253081:2257895 | 687 | 8 | + | 228 | 25738.3 | 6.53 | SVP |
| *PgMADS18* | Pg014451.1 | scaffold25:814656:818749 | 717 | 8 | − | 238 | 27124.6 | 9.52 | AGL6 |
| *PgMADS19* | Pg014630.1 | scaffold25:796521:803849 | 654 | 7 | + | 217 | 24695 | 7.93 | SOC1 |
| *PgMADS20* | Pg016044.1 | scaffold3:3612017:3628588 | 612 | 7 | − | 203 | 23056.1 | 4.99 | FLC |
| *PgMADS21* | Pg016673.1 | scaffold3:3637896:3641838 | 768 | 8 | − | 255 | 29008.8 | 8.85 | E(SEP) |
| *PgMADS22* | Pg017747.1 | scaffold33:1901905:1904548 | 759 | 8 | + | 252 | 28454 | 7.62 | AGL15 |
| *PgMADS23* | Pg018533.1 | scaffold37:1603815:1615806 | 837 | 7 | − | 278 | 31322 | 6.89 | FLC |
| *PgMADS24* | Pg020204.1 | scaffold40:966778:969371 | 699 | 7 | + | 232 | 26761 | 4.9 | B(AP3-PI) |
| *PgMADS25* | Pg020205.1 | scaffold40:971279:973422 | 708 | 7 | + | 235 | 27089.6 | 6.31 | B(AP3-PI) |
| *PgMADS26* | Pg022433.1 | scaffold5:1599117:1601542 | 675 | 7 | − | 224 | 25754 | 9.07 | B(AP3-PI) |
| *PgMADS27* | Pg023007.1 | scaffold50:465601:468168 | 750 | 7 | + | 249 | 28634.4 | 8.57 | E(SEP) |
| *PgMADS28* | Pg023008.1 | scaffold50:480602:487666 | 699 | 7 | + | 232 | 26732.3 | 9.43 | A(AP1-FUL) |
| *PgMADS29* | Pg024984.1 | scaffold60:1113804:1120065 | 753 | 8 | − | 250 | 28461.2 | 9.5 | E(SEP) |
| *PgMADS30* | Pg026995.1 | scaffold7:90410:99792 | 1974 | 11 | + | 657 | 74414.5 | 9.56 | C/D(AG) |
| *PgMADS31* | Pg027190.1 | scaffold71:566889:578438 | 654 | 7 | + | 217 | 24935.5 | 9.15 | SOC1 |
| *PgMADS32* | Pg027284.1 | scaffold72:255144:257303 | 528 | 6 | − | 175 | 20534.3 | 10.89 | TM8 |
| *PgMADS33* | Pg028066.1 | scaffold79:321847:329529 | 672 | 8 | − | 223 | 25020.5 | 10.39 | SVP |
| *PgMADS34* | Pg028180.1 | scaffold8:2827697:2833438 | 663 | 7 | − | 220 | 25172.5 | 10.04 | SOC1 |
| *PgMADS35* | Pg029246.1 | scaffold86:465224:468460 | 762 | 7 | − | 253 | 28671.7 | 8.89 | AGL17 |
| *PgMADS36* | Pg029338.1 | scaffold86:426938:437880 | 1443 | 13 | − | 480 | 55016.3 | 6.61 | AGL17 |

Note: CDS—coding sequence; MW—molecular weight.

The physicochemical properties of 36 MIKC-type MADS-box amino acid sequences were analyzed. The results were shown in Table 1. The length of the pomegranate MIKC-type genes varied greatly. The shortest length of the encoded protein was 167 aa, the longest was 657 aa. The corresponding protein molecular mass was between 19,349.0~74,414.5 Da, the average molecular weight was 28,553.23 Da. The isoelectric point (PI) ranged from 4.9 (*PgMADS24*) to 10.89 (*PgMADS32*), which contained nine acidic proteins and 27 basic proteins. These results would provide a theoretical basis for *PgMADS* protein purification and functional research.

### 3.2. Phylogenetic Analysis of Pomegranate MIKC-Type MADS-Box Family

To study the phylogenetic relationships in MIKC-type MADS-box proteins, these protein sequences of pomegranate, Arabidopsis and poplar were used to construct the phylogenetic tree (Figure 1). In this study, the phylogenetic tree was divided into 14 subfamilies, according to Díaz-Riquelme et al. [15] and Leseberg et al. [46]. Among the 36 type II members, only *PgMADS16* was classified as MIKC*. The other 35 members were classified as MIKC$^C$, whcih were divided into 13 subclasses. Among the 13 subfamilies, E (SEP) contained five members; SOC1, B (AP3-PI), and SVP each contained four members; C/D (AG), A (FUL) and FLC contained three member genes; AGL17 had two members. The group of AGL12, AGL6, BS, AGL15, and TM8 subfamilies had only one gene.

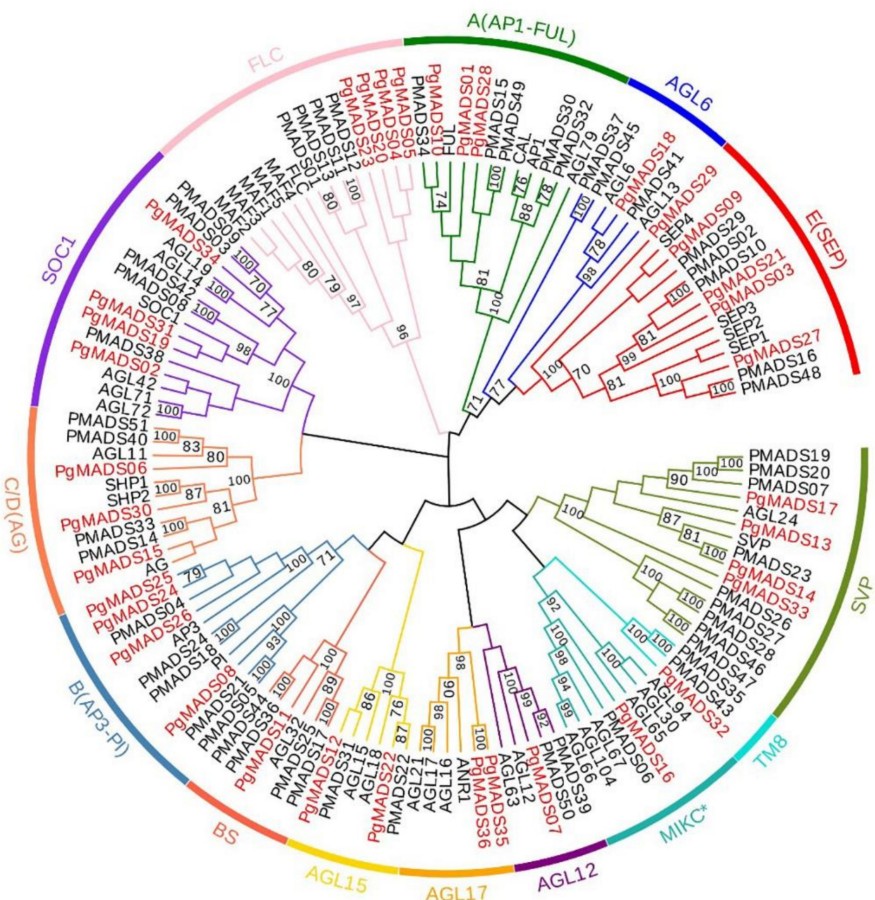

**Figure 1.** Phylogenetic tree of the MIKC-type MADS-box gene family in pomegranate (*PgMADS*), poplar (PMADS) and Arabidopsis.

### 3.3. Gene Structure Analysis and Motif Identification of Pomegranate MIKC-Type MADS-Box Family

In the prediction of conservative motifs of MIKC-type MADS-box members, 10 conservative motifs were identified in pomegranate (Figure 2). According to the distribution and number of motifs, the number of conservative motifs of MIKC-type MADS-box protein varied greatly in pomegranate.

Only *PgMADS16* contained two conservative motifs, while *PgMADS36* contained 12. The SMART analysis showed that motif 1 was the typical MADS-domain, consisting of about 60 amino acids, and motif 2 was located in the K-box domain. All *PgMAD*S proteins contain motif 1. Except for *PgMADS05*, *PgMADS20*, *PgMADS16*, and *PgMADS6*, other members contained motifs 3 with unknown functions. Besides *PgMADS28*, *PgMADS01*, *PgMADS04*, *PgMADS16*, and *PgMADS02*, other members contained conservative motif 4. Motif 5 only existed in A, E, AGL6 and AGL12 subfamilies. Motif 8 was shared by BS, SOC1, and AGL17. It was worth noting that the K-box domain was not found in *PgMADS16*, but there was a specific K-box motif 2.

The gene structures were investigated on the gene structure display server (GSDS). The length of genes ranged from 2 kb to 17 kb. Among 36 *PgMADSs*, there were 18 genes composed of 6 introns and seven exons, accounting for 50%. *PgMADS36* with 12 introns contains the most exons. Comparative analysis showed that the introns of TM8, BS, AGL15, and B subgroup genes were shorter, resulting in the shorter length of their genes than other subfamilies. Most FLC members had longer introns than other subfamily genes. The number of exons was seven in the B subfamily. The number of introns of the C/D class ranged from four (*PgMADS15*) to 10 (*PgMADS30*).

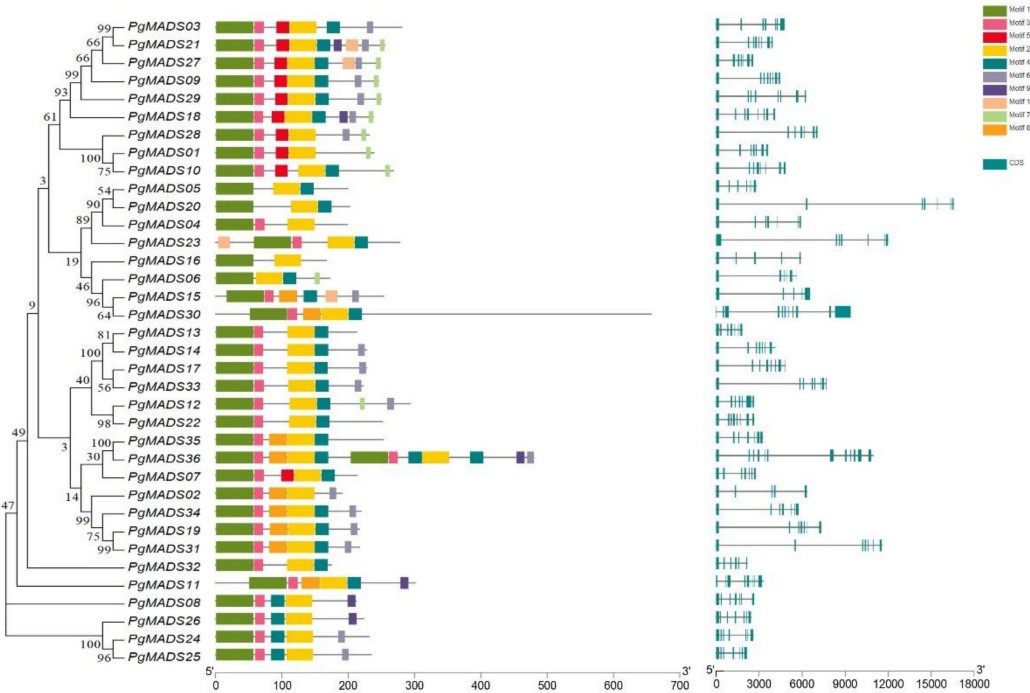

**Figure 2.** Phylogenetic tree, gene structure and conservative protein motifs of the MIKC-type MADS-box family in pomegranate.

### 3.4. Prediction Promoter Elements of the Pomegranate MIKC-Type MADS-Box Family

In this study, the *cis*-element analysis showed that promoters of *PgMADSs* contained a wide variety of *cis*-acting elements, ranging from 18 *cis*-acting elements in *PgMADS29* to 30 in *PgMADS30* (Figure 3). *PgMADSs* responded to different plant hormones and abiotic stress signals, including abscisic acid response elements (ABRE) [47], gibberellin response elements (GARE-motif, TATC-box and P-box) [48], auxin response elements (TGA-element, AuxRR-core, TGA-box and AuXRE) [49], methyl jasmonate response elements (TGACG-motif and CGTCA-motif) [50], salicylic acid response elements (TCA-element) [51], and stress-related response elements (ARE, TC-rich repeats, WUN-motif, LTR and MBS) [52]. The ABRE element was abundant in pomegranate, with 19 genes containing two or more ABRE. *PgMADS13* promoter sequence contained up to 10 ABRE, indicating a strong response to the abscisic acid signal. Many members contained gibberellin response elements, such as the

GARE-motif of *PgMADS01* in subfamily A, the TATC-box of *PgMADS09* in subfamily E, and the P-box of *PgMADS07* in subfamily AGL12. Except for AGL12 and TM8 subfamilies without LTR, other genes contained LTR elements. *PgMADS21* and *PgMADS35* contained two drought-induced MYB binding sites MBS. There were 15 genes containing auxin response elements, among which *PgMADS28* and *PgMADS12* contained two TGA-elements, while *PgMADS05* contained both TGA-element and AuxRE. *PgMADSs* were rich in methyl jasmonate and salicylic acid response elements. In addition to the AGL12 subfamily, the remaining genes of the AGL12 family contained one or more methyl jasmonate and salicylic acid response elements.

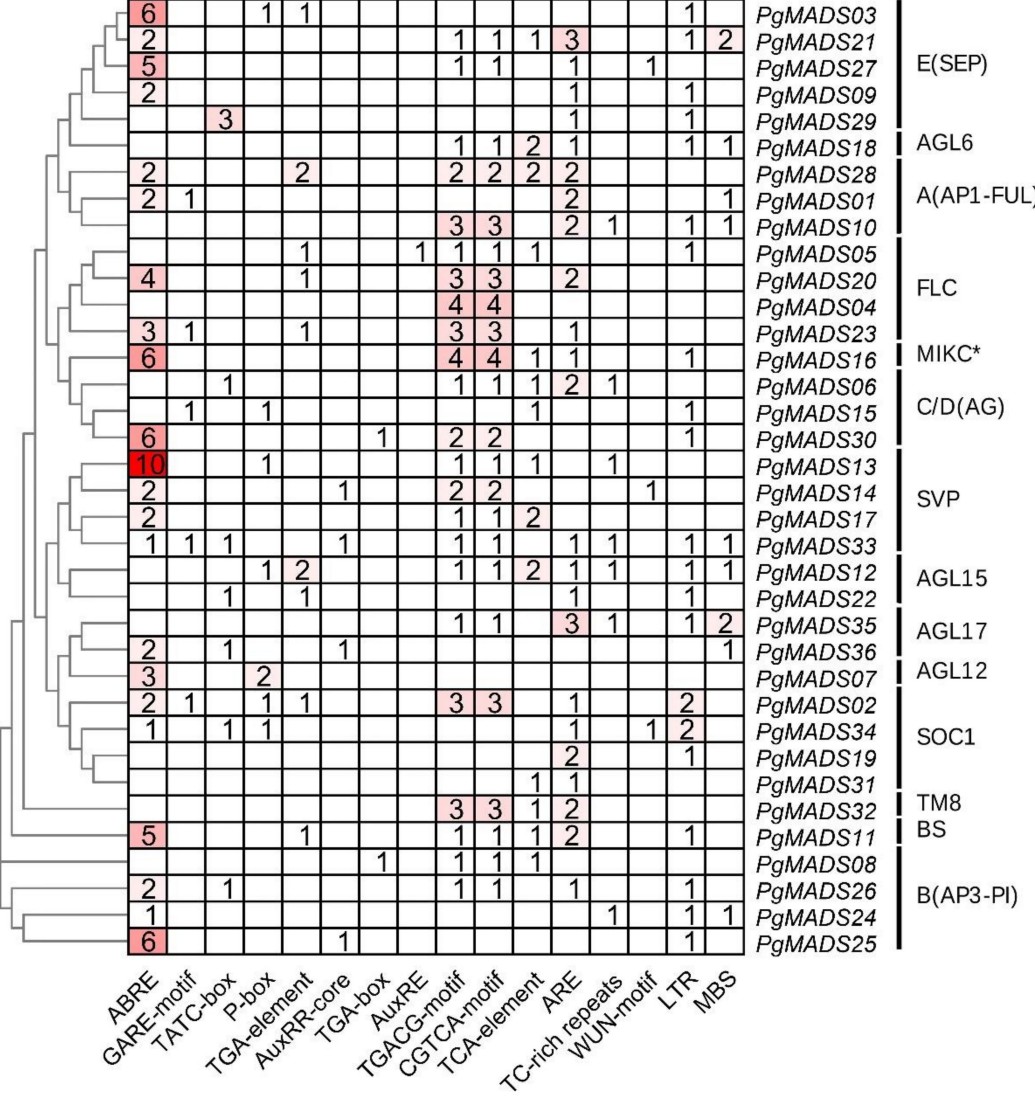

**Figure 3.** Analysis of *Cis*-acting elements of MIKC-type MADS-box gene family in pomegranate. ABRE is abscisic acid response element; P-box, TATC-box and GARE-motif are gibberellin response element; AuxRE and AuxRR are regulatory auxin responsiveness element; TGA-element and TGA-box are auxin response element; ARE is antioxidant response element; CGTCA-motif methyl jasmonate response element; LTR is cold stress response element; TCA-motif is salicylic acid response element; TGACG-motif is jasmonate response element. TC-rich repeats are cis-acting elements that respond to defense and stress responses; WUN-motif wound response elements; MBS is the drought resistance-induced MYB binding site.

### 3.5. Protein Interaction Networks of the Pomegranate MIKC-Type MADS-Box Family

The protein interaction between MIKC-type *PgMADS*s was analyzed to know potential functions and signal transduction or metabolic pathways. The result showed a close interaction among A, B, C/D, and E types, among which the C/D genes (*PgMADS30*, *PgMADS15* and *PgMADS06*) also had an interaction relationship (Figure 4). *PgMADS15* was located at the core of the interaction network and interacted with A, B, and E types genes. *PgMAD*S15 had stronger interaction with *PgMADS06*, *PgMADS09*, *PgMADS27*, and *PgMADS29*.

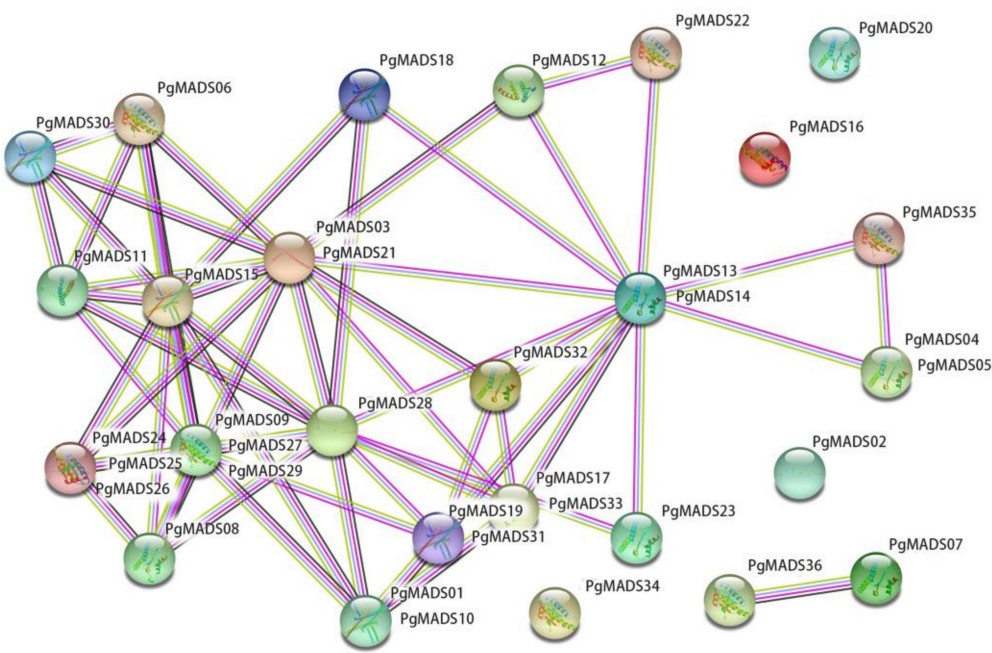

**Figure 4.** Protein-protein interaction network of MIKC-type MADS-box family in pomegranate.

Class B genes can form homologous dimers, but they steadily play function when they have formed heterodimers with AP3 and PI [53]. There was an interaction between AP3 (*PgMADS24*, *PgMADS25*, *PgMADS26*) and PI (*PgMADS08*) in pomegranate. E class genes could interact with multiple genes. When MIKC-type MADS-box protein formed a polymer, E genes acted as a 'binder' binds the dimer complex formed to form a polymer [25,26]. *PgMADS03* and *PgMADS21* belonging to E interacted with A, B and C/D classes genes, indicating that E genes might play an essential role in the formation of MIKC complexes related to the organ development of pomegranate flower.

According to the *PgMADSs* family protein interaction network, *PgMADS15* was at the core of the interaction network. The study further analyzed the protein network of *PgMADS15* in flower development, with the results shown in Figure 5. *PgMADS15* had high similarity with Arabidopsis AtAG (bit score 247.7, e-value 7.9E$^{-66}$). AG interacted with LFY (LEAFY) and AP2 proteins to regulate the flowering transition. AG, BEL1 and SUP coordinated to regulate ovule development. AG interacted with uncertain function proteins such as HUA1, ICU2, and LUG. It is predicted that *PgMADS15* had the function of regulating flowering, ovule, and other flower organ development.

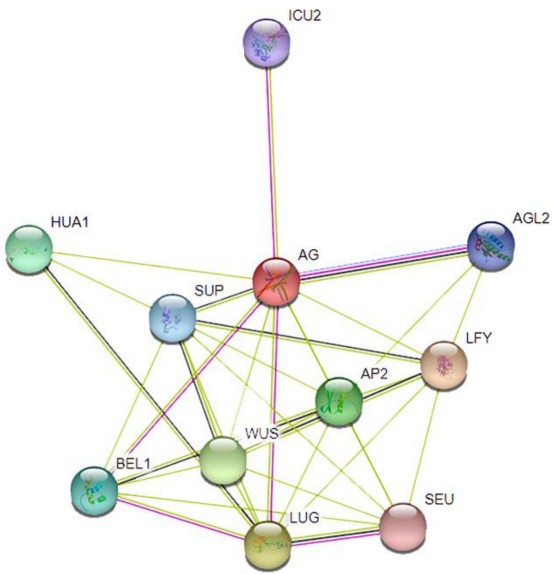

**Figure 5.** Protein interaction network of PgMADS15 in pomegranate.

*3.6. Expression Analysis for Pomegranate MIKC-Type MADS-Box Family*

According to the published transcriptome data, MIKC-type MADS-box genes are expressed in leaf, root, flower, fruit, endocarp, and pericarp (Figure 6). There were remarkable differences in tissue-specific expression and expression quantity. Significant differences were also found in the expression quantities of different subfamily genes of MIKC-type MADS-box (Figure 6a). The expression levels of E genes were the highest in flower organs, followed by C/D subfamily genes. Most genes of other subfamilies were low or not expressed in leaves and roots. The analysis found that the expression levels of the same subfamily genes in different tissues exhibited obvious differences, such as E-type genes. *PgMADS03* and *PgMADS21* were not expressed in root and leaf, while they were highly expressed in all tissues of pomegranate flower organs. *PgMADS09* was not expressed in leaves, roots, and inner/outer seed skins, while *PgMADS29* was almost not expressed in all tissues. These results indicated that the function of E genes might occur differentiation. *PgMADS06* and *PgMADS30* of C/D subfamily genes showed similar tissue-specific expression patterns, but *PgMADS15* was not expressed in functional male flowers and bisexual flowers. *PgMADS17*, *PgMADS19*, *PgMADS31*, and *PgMADS34* were only expressed in leaves and roots, indicating that the C/D subfamily genes of MADS-box expression were highly specific.

According to the differences in expression patterns, cluster analysis was carried out in the MIKC-type MADS-box family, with the results shown in Figure 6b. The obtained 36 genes can be divided into three groups. The genes in group I was highly expressed in flowers, mixtures, seed coats and peels, indicating that these genes play essential roles in the development of flowers and fruits. The expression levels of group II genes were lower than that in group I and III genes, and the genes of the second subgroup in group II were highly expressed in roots and leaves. The expression levels of group III genes were higher than in other groups in floral organ development. Cluster expression analysis showed that different subfamilies genes might have similar expression patterns, such as the E gene (*PgMADS03*, *PgMADS21*, and *PgMADS27*) and C/D gene (*PgMADS06* and *PgMADS30*). They were not expressed or behaved at a low expression level in roots and leaves, but were highly expressed in other tissues.

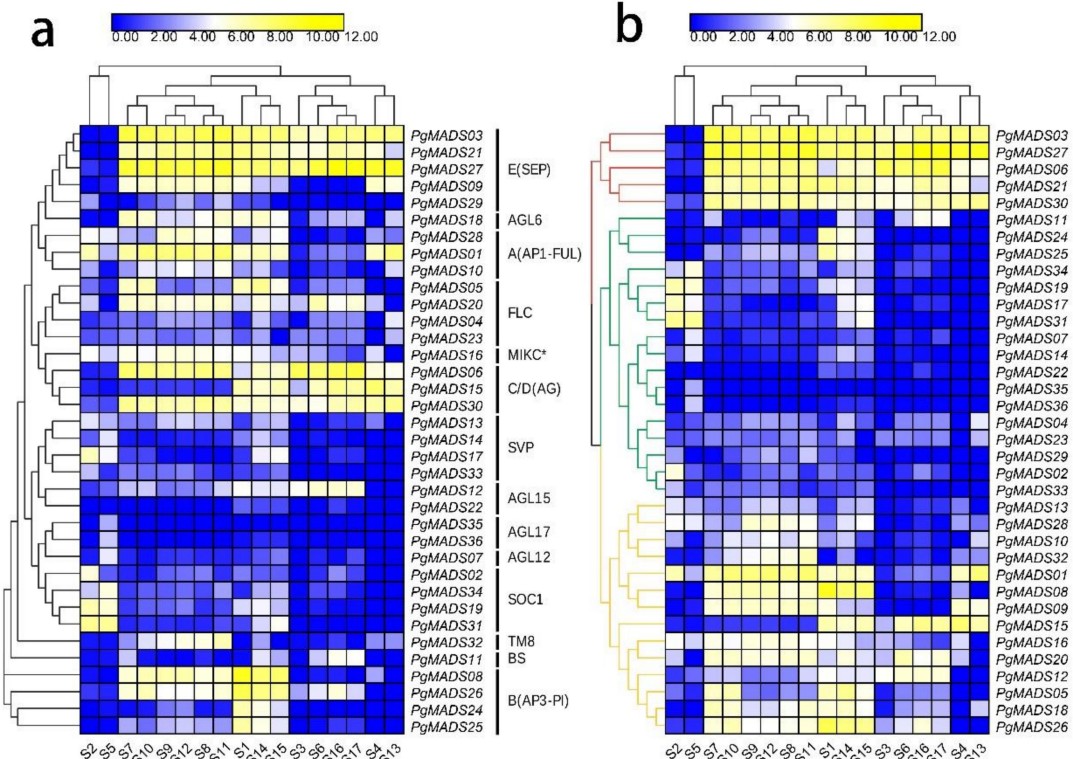

**Figure 6.** The heatmap of MIKC-type MADS-box family gene expression in different organs of pomegranate. Note: (**a**). Gene expression patterns of different subfamilies genes; (**b**). Hierarchical clustering of gene expression patterns. S1: flower; S2: leaf; S3: outer seed coat; S4: peel; S5: root; S6: inner seed coat; S7: functional male flower I; S8: functional male flower II; S9: functional male flower III; S10: bisexual Flower I; S11: bisexual Flower II; S12: bisexual Flower III; S13: 'Wonderful' peel; S14: a mixture of 'nana' flowers, leaves, fruits and roots; S15: a mixture of 'Black127' flowers, leaves, fruits and roots; S16: 'Baiyushizi' inner seed coat; S17: 'Tunisia' inner seed coat.

## 4. Discussion

In the plants, MIKC-type MADS-box transcription factors contain MADS-domain and K-box domain, and play important roles in flowering and floral organ development [54,55]. In recent years, the identification and evolutionary analysis of the MIKC-type MADS-box family have been reported in many species, such as 32 members in *Prunus mume* [13], 33 members in *Fragaria vesca* [56], 38 members in *Vitis vinifera* [15], 39 members in *Prunus persica* [57], 72 members in *Glycine max* [58], and 82 genes in *Malus domestica* [59]. Genomic data of three pomegranate varieties have been published on the NCBI database, but the identification and analysis of the MIKC-type MADS-box family have not been reported. In this study, a total of 36 candidate MIKC-type MADS-box genes were identified in pomegranate. Members were encoded with 167–657 amino acids. The 36 MIKC-type MADS-box members in pomegranate were far fewer than 201 in *Triticum aestivum* and 94 in *Nicotiana tabacum* [60,61]. The differences in the number of members may be due to the fact that tobacco and wheat are polyploid, while pomegranate being polyploid [29,62]. According to the phylogenetic relationships of MIKC-type MADS-box genes between Arabidopsis and poplar, *PgMADS* genes were divided into 13 distinct clades, including TM8 subfamily which was not found in Arabidopsis.

According to the analysis of gene structure, *PgMADS* genes contained 5–12 introns, and seven exons were the main structure of *PgMADSs*. The genetic structure of *PgMADSs* was consistent with that of Arabidopsis, apples, and soybeans [11,58,59], indicating that MIKC-type MADS-box gene structure is relatively conserved. Conservative sequence analysis of 36 MIKC-type MADS-box genes showed that members of the same subfamily had the same number of motifs and similar sequences, while there were some differences in the number of motifs between subfamilies. Also, members of the

same subfamily have certain similarities in gene structure, and these results support the findings of the subfamily classification of pomegranate MIKC-type MADS-box family. The conservative motif 1 encodes the MEF2-type MADS-domain, which is the most conserved domain. Motif 2 encodes the K-box domain, which is the second most conservative in *PgMADS* family. In general, the K-box motif exists only in the MIKC$^C$ subfamily [10], but there is motif 2 in MIKC* type *PgMADS16*. The gene structure of MIKC* and MIKC$^C$ family is similar. Mδ of type I genes is also treated as a type II gene in Arabidopsis and rice, so it is called the MIKC* type gene [12,63]. MIKC*-type genes share the characteristics of both M-type and MIKC$^C$ family. Furthermore, the evolutionary tree of Mδ is different from Mα, Mβ, and Mγ, but it is similar to that of MIKC-type [13,64]. It was speculated that the MIKC* subfamily might be the transitional form in the evolution of MADS-box genes. Xu et al. put forward a similar view that MIKC$^C$ subfamily lost K-domain, while retained more introns, making it became MIKC* subfamily [13]. With the further loss of introns, MIKC* subfamily became an M-type subfamily with shorter sequences and fewer introns [13].

*Cis*-acting elements exist in the gene promoters and form specific binding with transcription factors and play important roles in regulating the expression of target genes [65]. The promoter sequence of the MIKC-type MADS-box gene contains multiple *cis*-acting elements related to hormone response and abiotic stress, which are rich in methyl jasmonate response elements and abscisic acid response elements. This result is consistent with that by research reports on grape and *Phyllostachys edulis* [66,67], indicating that the promoter regulatory element of the MADS-box gene has a certain conservatism among different species. Previous studies have found that ABRE is associated with drought, ABA induction, and high salt stress in plants [68,69]. There were many ABRE elements in *PgMADS13*, *PgMADS03*, *PgMADS16*, *PgMADS30,* and *PgMADS25*, which were speculated to be related to pomegranate response to drought, ABA induction or high salt stress. MIKC-type MADS-box gene also contains other elements such as TGA-element, TATC-box, TGACC-motif, and TCA-element, indicating that the MIKC-type MADS-box gene can respond to the regulation of hormones such as auxin, gibberellin, jasmonic acid, and salicylic acid, which are involved in plant growth and development and plant stress response. These results suggest that MIKC-type MADS-box family genes may play essential roles in pomegranate's growth and development of pomegranate and in response to abiotic stress.

Studies have shown that the MADS-box gene play roles in the development of floral organs by forming homologous or heterologous complexes [70]. An analysis of the protein network of MIKC-type MADS-box family showed that there was the interaction among the MIKC-type MADS-box members, indicating that they might jointly regulate flower development by forming heterologous complexes. We also found that *PgMADS15* occupied a vital position in the protein interaction network, indicating *PgMADS15* played a crucial role in the regulation network of flower development.

Analysis of the expression pattern of MIKC-type MADS-Box gene found that *PgMADS17*, *PgMADS19*, *PgMADS31,* and *PgMADS34* were only expressed in leaves and roots, while other members were expressed in flower tissues. The results indicated that the MIKC-type MADS-box gene might be participated in pomegranate development regulation of different tissues. E subfamily genes (*PgMADS03*, *PgMADS21*, *PgMADS27*) and C/D subfamily genes (*PgMADS06* and *PgMADS30*) were highly expressed in various tissues of pomegranate flower organs, but not in leaves and roots, indicating that C/D and E subfamily genes were highly expressed specifically in pomegranate. Studies have shown that AP1-FUL, AP3-PI, AG, and AGL6 subfamilies genes regulate floral organ development in Arabidopsis, and the different subfamilies genes of MIKC-type MADS-box present similar expression patterns in different species [14,71,72]. AP1-FUL, AP3-PI, AG and AGL6 subfamilies genes have high expression levels in the pomegranate bisexual flowers, proving that these genes might be involved in the regulation of floral organ development in pomegranate. In *Arabidopsis thaliana*, the fruit differentiation and development are regulated by the C/D (*AGL11*, *SHP1*/*SHP2*) and A(AP1-ful) class genes [73,74]. Reports found that *FaMADS9*, a member of the strawberry E(*SEP*) class, was also involved in fruit development and maturation [75]. E class genes (*PgMADS03*, *PgMADS21*, *PgMADS27*, *PgMADS09*),

C/D class (*PgMADS06*, *PgMADS15*, *PgMADS30*), and A class (*PgMADS01*) were highly expressed in pomegranate peel, indicating that these genes might play important roles in pomegranate fruit development. The study showed that the same subfamily genes had different expression patterns, and the C/D genes regulated the development of carpel and ovule [21,76]. In pomegranate, the C gene *PgMADS06* and D gene *PgMADS30* were highly expressed at three development stages of functional male and bisexual flower, while the D gene *PgMADS15* was not expressed, suggesting that *PgMADS15* may lose some functions in the evolution and development of species.

## 5. Conclusions

Herein, 36 MIKC-type MADS-box genes were identified in pomegranate, and the necessary information concerning these genes was analyzed by bioinformatics to explore their phylogenetic relationship and their functions in the development of flower organs. The gene structure and conserved domain of 36 *PgMADSs* were analyzed, showing that the same subfamilies' gene structures were relatively similar. *Cis*-acting element analysis showed that promoter sequences of *PgMADS* genes contained multiple hormone response and abiotic stress related elements, suggesting that *PgMADSs* might be closely related to plant hormone signal transduction and adverse situations. It was speculated that *PgMADSs* might be related to the growth and development of pomegranate. Protein interaction network analysis found that *PgMADS15* was located at the core of the interaction network. Combining the analysis of tissue specific expressions, *PgMADS15* was highly expressed in the peel and inner seed coat. It was concluded that *PgMADS15* might play an essential role in the peel and inner seed coat development of pomegranate. Our conclusions will lay the foundation for further research on the function of the MIKC-type MADS-box gene in pomegranate, primarily providing a reference for the analysis of the gene function regulating the development of flower organs.

**Supplementary Materials:** The following are available online at http://www.mdpi.com/2073-4395/10/8/1197/s1, Table S1: RNA-Seq data of pomegranate.

**Author Contributions:** Conceptualization, Y.Z. and H.Z.; methodology, Y.Z. and H.Z.; investigation, Y.Z. and H.Z.; writing—original draft preparation, Y.Z. and H.Z.; writing—review and editing, Y.Z., Y.W., X.Z. (Xinhui Zhang), X.Z. (Xueqing Zhao), and Z.Y.; supervision, Z.Y.; funding acquisition, Z.Y. All authors have read and agreed to the published version of the manuscript.

**Funding:** This research was funded by the Initiative Project for Talents of Nanjing Forestry University [GXL2014070, GXL2018032], the Priority Academic Program Development of Jiangsu High Education Institutions [PAPD], the National Natural Science Foundation of China (31901341), and the Natural Science Foundation of Jiangsu Province [BK20180768].

**Acknowledgments:** Y.Z. and H.Z. are co-first authors.

**Conflicts of Interest:** The authors declare no conflict of interest.

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
