# Peer review of "Genome-Wide Identification and Expression Analysis of MIKC-Type MADS-Box Gene Family in Punica granatum L."

_agronomy, doi:10.3390/agronomy10081197_

Round 1

Reviewer 1 Report

Zhao et al. explored the "Genome-wide identification and expression analysis of MIKC-type MADS box proteins in P. granatum L". This work has merit and can be of interest to the audience of Agronomy upon publication. However, I have some concerns about the manuscript.

Major comments:

  1. Please mention about how you classified these 36 genes into families, subfamilies and clade. This will help to understand about your paper at a glance.
  2. Line 156-160: How do you classify M type and MIKC type family to obtain 36 MIKC type MADS box members?
  3. Line 162-163: How do you rename MADS box proteins? Do you follow any nomenclature? Do you update the renamed genes in the public database?
  4. I recommend to add columns about clade type and families and subfamilies in Table 1
  5. How you do divide the genes into clade, family and subfamily. Please write down the rules and nomenclature in the Methods section.
  6. For validating the results from insilico, I recommend the gene expressions studies for some set of genes from each family. Without validations, the results cannot sound scientific.
  7. In line 265: Please add published transcriptome reference.

Other comments

  1. Please write all genes in italics
  2. Line 303: Please check the Glycine max spelling
  3. Similarly, please check spelling of MIKC in line 315. Many spelling errors throughout the manuscript.
  4. Please check for the reference format according to the journal format.

Reviewer 2 Report

Dear Authors,

Yujie Zhao, Honglian Zhao, Yuying Wang, Xinhui Zhang, Xueqing Zhao and Zhaohe Yuan: Genome-Wide Identification and Expression Analysis of MIKC-Type MADS-box Gene Family in Punica granatum L.

In this study, bioinformatics methods were used to identify 36 MIKC-type MADS-box members, and their gene structure, cis-acting elements and tissue-specific expression were analyzed to explore their functions in the development of pomegranate (Punica granatum L.) flowers. Gene structure and conserved domain analyzation showed that the same subfamilies’ gene structures were relatively similar. Cis-acting element analysis indicated that promoter sequences of PgMADS genes contained multiple hormone response and abiotic stress related elements, suggesting that PgMADSs might be closely related to plant hormone signal transduction and adverse situations. Protein interaction network analysis found that PgMADS15 was located at the core of the interaction network. PgMADS15 was highly expressed in the peel and inner seed coat. It was concluded that PgMADS15 might play an essential role in the peel and inner seed coat development of pomegranate. Results of this study will provide a reference for the classification, cloning and functional research of pomegranate MADS-box genes.

The manuscript is written well, congratulation for the authors! I recommend to accept the MS with minor revision, my excuses you can find in detailed below.

The title and abstract (less then 200 words) are OK.

Introduction: is also OK, except some notices:

lines 49-51:As transcription factors, MADS-box genes involve in

many processes, such as floral meristem development, flowering time, floral organ determination and development, and seed development and maturation.” This is the repetition of the lines 40-43, and therefore you can delete here.

line 83: The citation of the Ref 28. in line 471 is not correct because it is a book citation, Please, follow the Instructions for Authors  in Agronomy to correct this like this:  

Books and Book Chapters:
 Author 1, A.; Author 2, B. Book Title, 3rd ed.; Publisher: Publisher Location, Country, Year; pp. 154–196.

An other note that in the References you shall have to correct all the cited papers like below: especially taking care for the Volume which shall have to be in italics always according to the Instructions for Authors in Agronomy.

Author 1, A.B.; Author 2, C.D. Title of the article. Abbreviated Journal Name Year, Volume, page range.

Mat&Methods:

line 109 (and everywhere when it appears): the „poplar” is not  a species name, so it is written in small letter

line 117: squared parenthesis for Refs. 45 and 51.

line 148: „Through Kallisto 0.44.0 software [36]” – here you shall have to mention a website for the software also.

Results:

line 172: the Table 1 title you should put under the table (into line 174) and you should give a more informativ title for this table (referring to what does it contain).

line 186: ): the „pomegranate and poplar” words are not species names, so they shall have to write in small letters (please, correct everywhere in the text).

lines 213-223: here you mention that pomegranate  genes  have several cis-response elements like from ABRE till MBS. Here I miss the references for the response elements, please, replace them.

lines 234-235: in the Figure 3. legend, you mention the CAT/CAAT-box elements, but they are not exist in Fig.3. This shall have to correct anyhow.

lines 253-269: this paragraph you shall have to put under the Figure 4. (under line 261) since here you speak about the specific role of PgMADS15 protein. In line 255, you shall have to change the Figure 6. for Figure 5. because this latter is the correct nomination.

Discussion:

lines 309- 310: „The difference in the number of members may be caused by tobacco and wheat being polyploid and pomegranate being polyploid [46].”

In Ref. 29 (Qin et al. 2017) what you cited for sequencing of pomegranate: „ The genome of an elite pomegranate cultivar ’Dabenzi’ (2= 2= 18)….. was sequenced”.  Dabenzi cultivar was use by these authors and you also which seem to be diploid according to Qin et al. 2017.

and according to Ref. 46. in your References:

Xu, Y.; Ding, Z.; Yao, Y.; Gong, R.; Wen, Y. Karyotype analysis of four pomegranate cultivars. Nonwood Forest Research 2008, 26, 47-52.

„The results showed that all 4 cultivars had 18 chromosomes and the karyotype belonged to stebbins' 2B type.”

May be you wanted to say here that the the pomegranate is diploid therefore there are less MICK-type  MADS-boksz members in pomegranate as in e.g. the wheat?!..

line 311 and line 313: „PgMADS genes „ shall have to put into italics

lines 315- 316: „indicating that MICK-type MADS-box gene structure is relatively conservation conserved.”

line 330: Xu et al. is the Ref.13, not the Ref 12.

line 332. like above, it is may be the Ref.13. instead of Ref.12.

line 403: „Acknowledgments: Y.Z. and H.Z. are co-first authors.” I guess this information you shall have to write under the Author List and Affiliations with using a star symbol for both authors.

General note: please check in the MS that the gene names shall have to write in italics.

That is all. Have a nice day!

Sincerely yours,

Reviewer 1

Round 2

Reviewer 1 Report

The manuscript has improved drastically. Hence, now it can be published in the Agronomy.

Author Response

Thanks reviewer for your recognition and affirmation of our work.